# Coherent Diffraction Imaging in Transmission Electron Microscopy for Atomic Resolution Quantitative Studies of the Matter

**DOI:** 10.3390/ma11112323

**Published:** 2018-11-19

**Authors:** Elvio Carlino, Francesco Scattarella, Liberato De Caro, Cinzia Giannini, Dritan Siliqi, Alessandro Colombo, Davide Emilio Galli

**Affiliations:** 1Istituto per la Microelettronica e i Microsistemi, Consiglio Nazionale delle Ricerche (CNR-IMM), Sezione di Lecce, Campus Universitario, via per Monteroni, 73100 Lecce, Italy; francesco.scattarella@gmail.com; 2Istituto di Cristallografia, Consiglio Nazionale delle Ricerche (IC-CNR), via Amendola 122/O, 70125 Bari, Italy; liberato.decaro@ic.cnr.it (L.D.C.); cinzia.giannini@ic.cnr.it (C.G.); dritan.siliqi@ic.cnr.it (D.S.); 3Dipartimento di Fisica “Aldo Pontremoli”, Università degli Studi di Milano, via Giovanni Celoria 16, 20133 Milano, Italy; alessandro.colombo6@unimi.it (A.C.); davide.galli@unimi.it (D.E.G.)

**Keywords:** TEM, electron diffraction, electron coherent diffraction imaging, atomic resolution imaging, phase retrieval

## Abstract

The paper focuses on the development of electron coherent diffraction imaging in transmission electron microscopy, made in the, approximately, last ten years in our collaborative research group, to study the properties of materials at atomic resolution, overcoming the limitations due to the aberrations of the electron lenses and obtaining atomic resolution images, in which the distribution of the maxima is directly related to the specimen atomic potentials projected onto the microscope image detector. Here, it is shown how augmented coherent diffraction imaging makes it possible to achieve quantitative atomic resolution maps of the specimen atomic species, even in the presence of low atomic number atoms within a crystal matrix containing heavy atoms. This aim is achieved by: (i) tailoring the experimental set-up, (ii) improving the experimental data by properly treating parasitic diffused intensities to maximize the measure of the significant information, (iii) developing efficient methods to merge the information acquired in both direct and reciprocal spaces, (iv) treating the dynamical diffused intensities to accurately measure the specimen projected potentials, (v) improving the phase retrieval algorithms to better explore the space of solutions. Finally, some of the future perspectives of coherent diffraction imaging in a transmission electron microscope are given.

## 1. Introduction

High Resolution Transmission Electron Microscopy (HRTEM) makes it possible to image the interference pattern transmitted by a thin specimen illuminated by an electron wave-field [1]. The pattern intensity distribution is a function of the specimen projected-potential and depends on the used electron optical conditions [1]. At the objective lens optimum defocus, and if the specimen is thin enough to satisfy the approximation of weak phase object [1], the distribution of the minima in the HRTEM image corresponds to the distribution of the specimen atomic column projected-potentials, carrying important information on the structural properties of the sample at atomic resolution [1]. In general, this is not straightforward and it is necessary to simulate the HRTEM image intensity distribution by appropriate codes and tentative structural models to derive reliable and accurate structural information [1]. The image resolution is lens aberrations limited, worsening resolution of about two orders of magnitude above the diffraction limit threshold [1]. Recently, the development of spherical and chromatic aberrations correctors made it possible to achieve a resolution of about 25λ [2], which is a big leap in the capability to study the properties of the matter at high resolution and accuracy by HRTEM. Moreover, the limitation imposed by the lens aberrations can also be overcome by physical methods enabling to derive the wave scattered by the specimen at a resolution in principle limited only by the diffraction threshold. Electron holography [3], exit phase reconstruction [4] and coherent diffraction imaging [5,6,7] were developed for these purposes. In particular, by using non-aberration corrected equipment, electron diffraction imaging (EDI) has reached a resolution better than 30λ, revealing basic properties of the matter, even in the presence of light atoms, not detectable by the relevant standard HRTEM experiments [7]. It should be pointed out that the use of aberration correctors or of methods to achieve the right diffused wave-functions, despite the lens aberrations, are not antagonists. Indeed, the application of the latter to experiments performed by aberration corrected equipment would facilitate studying the matter with higher resolution and accuracy.

In the EDI approach, an image of a specimen illuminated by a coherent wave field is obtained by recording the wave field diffused intensity and retrieving the relevant phase; the latter being not directly measurable in the diffraction experiment [5]. This has been experimentally demonstrated for the first time by using a coherent beam of X-rays in a synchrotron beam line [8]. In 2003, this approach was tailored for the earliest coherent diffraction imaging experiment in a TEM [5]. The first coherent diffraction imaging experiments in TEM were achieved on single nanoparticles [5]. In 2012 the approach was also demonstrated for extended specimen by using a self-confined illumination, giving rise to the so-called keyhole electron diffraction imaging (KEDI) [9].

Here, we review the steps necessary to perform an EDI/KEDI experiment in TEM, with the help of some examples, discussing the basics behind each of them and the relevant bibliography. We also developed for the first time what we termed augmented EDI, namely the improvement of imaging results applying a priori information to the experimental data to boost the information that can be extracted from them.
-In Section 2, the EDI/KEDI experimental set up is introduced. Experimentally, EDI/KEDI require the acquisition of one HRTEM image and one nanodiffraction pattern, by using the same illumination configuration on the same specimen area [7]. The need to fulfill the theoretical requirement necessary for a successful EDI/KEDI experiment poses some constraints on the experimental conditions and on the specimen illuminated area. The basis of the theory for phase reconstruction will be introduced, accounting for the Shannon theorem [10] and the generalized sampling theorem [11].-In Section 3, the data reduction strategies are discussed. The experimental images and nanodiffraction patterns need an appropriate data reduction prior to apply the phasing procedure. As will be shown, the data reduction can also involve methods to maximize the information that can be extracted from EDI/KEDI experiments resulting in an augmented EDI/KEDI quantitative imaging [12,13].-In Section 4, the phase retrieval processes are discussed. The phasing process requires algorithms capable of retrieving the phase lost in the nanodiffraction experiments. Different kinds of phase retrieval algorithms will be applied. In particular, we focus on recent developments capable of minimizing the reconstruction errors starting from low-resolution information of the support and from completely random phases [14].-In Section 5, we draw the conclusions and discuss the future perspectives of EDI/KEDI.


## 2. EDI/KEDI Experimental

All the experiments in what follows were performed at room temperature by using a Jeol (JEOL Ltd., Tokyo, Japan) JEM 2010 TEM/STEM microscope operated at 200 kV (electron wavelength λ = 2.51 pm). The microscope has a low spherical aberration coefficient objective pole piece (Cs = 0.47 ± 0.01 mm) and no correctors of spherical and chromatic aberrations. The measured resolution at optimum defocus in phase contrast HRTEM is of 0.19 nm and the equipment has, in Scanning TEM (STEM) high angle annular dark field imaging, a resolution of 0.126 nm. The microscope is located in a stable environment with thermal drift <0.1 Celsius/h and mechanical vibrations and external magnetic stray fields one order of magnitude below the specifications for this class of microscopes. The images and nanodiffraction patterns were acquired on a standard phosphorous scintillator coupled to a 1024 × 1024 Charge Coupled Device (CCD, mod. MSC496, Gatan Inc., Pleaseton, CA, USA). The physical size of each detector bit is of 19 µm.

Let us consider an object illuminated by wave front: the specimen scattering function is proportional to the inverse Fourier transform of the wave diffused by the sample [1]. The diffused wave function has two components: the amplitude and the phase. In diffraction experiments the measured data are the intensities of the scattered waves, as the phase information is completely lost [15]. This is the phase-problem, well known since the beginning of the diffraction experiments in the first years of the last century. The Shannon’s theorem [10] states that “*any function f(x,y) non-zero in an interval S, can be fully reconstructed by sampling its Fourier transform at the Nyquist frequency 1/S*”. This theorem hence paves the way for the reconstruction of the phase lost in a diffraction experiments, as recognized by David Sayre [16] few years after the publication of Shannon. The phase reconstruction problem is a typical ill-posed mathematical problem that, to be solved, needs the use of a priori information. Furthermore, the information in the diffraction pattern is affected by experimental uncertainties, due to the eventual presence of a beam stopper (which makes the pattern incomplete), to the drift of the specimen, to inelastic scattering contributions, to the poor signal to noise ratio of the measurement, to the limits in the detectors due to the poor dynamic range or in defects of some pixels, etc. The shape of the object to be reconstructed is a common a priori information that makes it possible to successfully reconstruct the phase [8]. This information is readably accessible in EDI/KEDI experiments in TEM, by using the standard HRTEM image information on the shape of the scattering object (see graphical abstract for the scheme of a coherent diffraction imaging experiment in TEM).

The way practically used in many cases to recover the phase, is by using iterative Fourier-transform algorithms [17] based on the Gerchberg–Saxton algorithm [18]. The accuracy and the precision in the phase reconstruction and imaging by using iterative Fourier-transform algorithms are based on the robustness of the available information. The structural information contained in the intensity I(***k***) of an electron diffraction pattern of a thin specimen illuminated by a wave function Ψ(***r***) with wavelength λ is described in the kinematical approximation [5] by:(1)$$I\left(k\right)\propto ∥\overset{+\infty }{\underset{-\infty }{\int }}\psi \left(r\right)\left[1\;+\;i\pi \lambda U\left(r\right)\right]\exp\left(2\pi ik\cdot r\right)dr{∥}^{2}\;$$
where $U\left(r\right)\;=\;-2meV\left(r\right)/h^{2}$, *V*(***r***) is the Coulomb potential of the sample medium, *m* and *e* are the mass and the charge of the electron respectively, and *h* is the Planck’s constant. For plane wave illumination, the electron diffraction intensity is related to the Fourier transform *F*(***k***) of the potential *U*(***r***) by [5]:(2)$$I\left(k\right)\approx \delta \left(k\right)+{\left(\pi \lambda \right)}^{2}F^{2}\left(k\right)$$

The resolution of the reconstructed image is related to the higher spatial frequency measurable in the diffraction pattern. Indeed, there are two terms in the resolution: one term is due to the periodic component, related to the diffracted Bragg’s intensity if the specimen is crystalline [9], and the other is due to the non-periodic component, i.e., the shape of the object [19]. In the latter case, from the Shannon’s theorem, the minimal spatial frequency required to sample a finite object of size *S* in the reciprocal space is 1/*S*. *S* is the so-called support, which is the finite region of non-zero scattering illuminated by the coherent probe [17]. If *f* is the smallest frequency measurable in the diffraction pattern, hence ***Sf* ≥** 1. (***Sf***)***^−n^***, with *n* the dimension of the image, is the oversampling ratio [8] and mathematically it must be at least 2 to tackle the phase retrieval process successfully [20]. This requirement and the coherent length of the electron sources nowadays available pose a constraint on the linear size of the object to be imaged by electron coherent diffraction imaging to about 10–20 nm [19]. The whole illuminated area from which the HRTEM image and the nanodiffraction patterns are acquired has a diameter of the order of about 50 nm, to satisfy the relevant oversampling requirements. An example of EDI experiment is reported in Figure 1. In this case the rod is about 18 nm in length and 5 nm wide, whereas the illuminated area is of 40 nm [7] (not all the illuminated area is shown in the picture).

Figure 1 shows the peculiar steps of an EDI experiment: Figure 1a is the HRTEM image of a nanorod of TiO_2_, oriented along the [100] zone axis, at a resolution limited by the electron lens aberrations to 190 pm [7]. The HRTEM image is used: (i) to estimate the size and shape of the rod to be used as a priori information in the phase retrieval process; (ii) to estimate, by its Fast Fourier Transform (FFT), some of the low frequency diffracted intensities lost in the nanodiffraction pattern due to the beam stopper. Figure 1b reports the nanodiffraction pattern, complete with direct beam and some of the low frequency intensities, as derived from the FFT of the HRTEM after proper rotation and scaling [5,7]. Figure 1c shows the phase reconstructed image at a resolution of 70 pm [7], which makes it possible to distinguish the projection of the oxygen atomic columns on the TiO_2_ anatase (100) crystallographic plane (see the anatase projected crystal cell in the inset). It is worthwhile to remark, that the oxygen atomic columns closer to the Ti atomic columns are invisible in the magnified HRTEM image in Figure 1d, due to the limited resolution caused by the electron lenses aberrations. The about threefold gain in resolution enables to detect the distortion of the TiO_2_ anatase crystal cell, related to the photocatalytic properties of TiO_2_ anatase at the nanoscale [7]. 

In KEDI experiments, an extended crystalline specimen can be studied across smaller areas confining the electron probe by using the condenser lenses of the microscope. This is the way, for example, to study the detail of an interface between two materials or a crystal defect such as a dislocation or a stacking fault. Also, in KEDI experiments there are some constraints to be fulfilled due to theoretical requirements of the approach. In this case the resolution ρ achievable in the reconstructed image is related to the size of the illumination function [9], which is the support for KEDI. In addition, it is experimentally required to collect an HRTEM image and a nanodiffraction pattern from the same specimen area and by using the same electron illumination conditions. The cathode emission condition and the electron optical illumination system of the microscope have to be set up in free lens control to increase the probe coherence on the smallest illuminated area achievable. The microscope has to be operated starting from a standard nano-probe configuration with the smallest condenser aperture, typically a condenser aperture of 10 micron. The three magnetic lenses of the illumination system of the microscope have to be operated independently, together with the electrostatic lens of the emitter, to produce the smallest probe on the focal plane of the pre-field of the objective lens to obtain the smallest coherent parallel beam on the specimen. To further increase the coherence of the electron-probe, the heating current has to be decreased with respect to the heating current used in standard TEM imaging. Typically, this cathode set-up results in a reduction of the emission current of about 50% in a probe which has a minimum size on the specimen between 5 nm to 10 nm. The current density on the specimen is below the detection limit of the microscope phosphorus screen (<0.1 pA·cm^−2^). This set up makes it possible to acquire the diffraction pattern without using the beam stopper [9]. This is an immediate advantage with respect to the experimental set up used in EDI where the use of the beam stopper makes the diffraction pattern incomplete. In KEDI experiments, to achieve a resolution ρ, it is necessary to measure in the reciprocal space at least a frequency of (1/ρ) pm^−1^, and the pixel size of the detector should be at least two times smaller than ρ to produce an image not pixelated. For example, to achieve an image resolution of 70 pm, using a 1024 × 1024 array detector, with physical pixel size of about 19 microns, the detector has a total field of view (FoV) of about 30 nm. To achieve the Nyquist’s oversampling conditions the area of the specimen illuminated by the electron beam has to be $\leq 2^{-\frac{1}{2}}\cdot \left(\mathrm{FoV}\right)≅21\;\mathrm{nm}$. In Figure 2, an example of KEDI experiment, applied to a bulk specimen of Si oriented along the [112] direction, is shown. A silicon specimen in [112] orientation represents a benchmark for the spatial resolution as in this configuration the Si atoms in the dumbbell are 78 pm apart. The HRTEM image in Figure 2a shows the self-confined illuminated area, which represents the support in KEDI experiments. The interference fringes for Si (112) are visible within the illuminated area, but the resolution, aberration-limited at 190 pm at optimum defocus [1], prevents to resolve finer details of the interference pattern. The size of the illuminated area is of about 9 nm and hence the conditions for resolving the Si dumbbell at 78 pm in KEDI are largely satisfied. In Figure 2b the nanodiffraction pattern is reported; the arrows point the highest spatial frequency reflections measured, corresponding to a spacing of 72 pm; hence the highest spatial frequency information is enough to resolve the Si dumbbell. In Figure 2c, the image after phase recovery is reported; the red square marks the region shown magnified in Figure 2d. For reader convenience, in the top left part of Figure 2d, the Si crystal cell (in blue) and the Si atomic column positions (in red), projected on the (112) crystal plane, are reported. In the inset below the red line, the simulation of the Si (112) projected potential is reported. The experimental intensity profile along the red line in Figure 2d is shown in Figure 2e (solid line) together with the relevant simulation of the Si (112) (red dots) projected potential intensity profile. As expected, the resolution of the phase recovered image enables to safely achieve the resolution of 78 pm, necessary to distinguish the Si atoms in the dumbbells. Some small artifacts at low intensities, in between the projected atomic columns, are visible in Figure 2d. These artifacts are similar to those shown in Figure 6 of Ref. [9] and could be related to small dynamical diffraction contributions to the diffracted intensities.

The peculiarity of KEDI experiments on crystalline specimen, is to have nanodiffraction patterns where most of the diffracted intensities are concentrated in the Bragg spots, whereas the intensities in between the spots are extremely weak and, in most of the cases, buried in the experimental noise.

In this respect, the KEDI experiments are particularly *ill posed* problems. Nevertheless, numerical simulations show that, even in the extreme case in which 99% of the diffracted intensities between the Bragg spots are unknown, the a priori information available from the HRTEM image, even if at a resolution limited by the aberrations, are sufficient to correctly retrieve the phase of the diffracted wave [9]. In these cases, the solution to the phase problem is related to the generalized sampling theorem [11] that marks how the useful knowledge for a correct sampling of the function to be reconstructed is not only the sampling of function itself, but also of its functional as, for example, in the case of its derivative [9]. The sampling of the function and of its functional generates a number of independent equations capable to overcome the lack of information between the Bragg spots of the function to be reconstructed [9,21].

## 3. Data Reduction

An important part of a successful EDI/KEDI experiment regards the treatment of the experimental data before the application of the phase retrieval algorithms [19]. Since the first works on coherent diffraction imaging in TEM, it was evident that both HRTEM image and nanodiffraction need processing before phase retrieval [5]. The basic data reduction for nanodiffraction patterns includes background subtraction, noise reduction and eventual masking whereas, for the relevant HRTEM images, it is necessary to apply an accurate process of scaling and rotation in order to have the relevant FFT that matches precisely the nanodiffraction pattern. Furthermore, in many cases, due to instrumental limitations, the nanodiffraction patterns in EDI/KEDI experiments are affected by parasitic intensities that complicate the measure of the genuine diffracted intensities and the application of an efficient background subtraction. In fact, the thresholding process prior of background subtraction can cancel, in many cases, most of the weak diffracted intensities [13]. Actually, the Bragg spots at higher frequency, which carry the information at higher spatial resolution, can be very weak and can have intensities comparable with the parasitic intensities. To measure all diffracted intensities with accuracy it is convenient, if not mandatory, to remove the parasitic intensities to obtain a diffraction pattern where all the remaining intensities are significant for the phase retrieval process. Finally, thresholding of the resulting pattern for background subtraction can be more efficiently and safely performed. The parasitic intensities, in the nanodiffraction pattern for EDI/KEDI, have a variety of different physical origins and can be removed without affecting the true signal. In this way, the resulting diffraction pattern is suitable for further process to maximize the signal to noise ratio and the information that can be extracted by the phase retrieval process [13]. In this respect, it is noteworthy that the relationship between the projected potential and the diffracted intensity has the simple expression reported in Equation (2) only if the specimen is thin enough to satisfy the kinematical approximation [22]. Strictly speaking, the specimen-electrons interaction in a TEM is such that the kinematical approximation is never completely satisfied and the electrons can experience several scattering events during the propagation along the specimen, resulting in diffracted intensities that cannot be described by the kinematical approximation [22]. This is the physical reason why the methods usually effective in X-ray crystallography cannot be straightforwardly applied in electron crystallography [15]. This is also the reason why EDI/KEDI experiments need to be performed on thin specimens. Indeed, if the specimen thickness is relatively high, it can produce subtle artifacts [9]. For example, the KEDI experiment reported in Figure 2 has been performed on a Si specimen 6 nm thick, resulting in a successful imaging at 70 pm of spatial resolution. Similar experiments, performed on Si specimen 12 nm thick, produced unreliable Si (112) atomic positions [9] in the phase reconstructed image. It is noteworthy, that a priori information can be used to correct the dynamical diffraction component of the pattern in the case of dynamical diffraction patterns in EDI/KEDI experiments. This is in analogy with what is done for *ill posed* problems, where a priori information is necessary to solve the system of equations. The use of a priori information enables to correct the dynamical component of the patterns, resulting in augmented EDI/KEDI images, which are quantitative structural map of the atomic projected potential of the specimen [12]. To achieve this result, a priori information of the chemistry of the specimen, known by other routes or measurable by energy dispersive X-ray spectroscopy or electron energy loss spectroscopy in the same TEM session, or information on the structure of the specimen, can be used to scale the diffracted intensity by imposing loose mathematical constraints to the measured diffracted intensities. This issue has been recently tackled by De Caro et al. [12]. An example is shown in Figure 3 on a case study of SrTiO_3_ specimen in [001] orientation.

Figure 3 on the left, is the HRTEM image of a KEDI experiment used to derive the support. Figure 3a is the relevant raw nanodiffraction pattern. In the left inset of the HRTEM, the lattice interference fringes are shown at higher magnification; in the right inset, the relevant image simulation is reported. The latter also shows the expected positions of the SrTiO_3_ atoms, in [001] projection, by dots of different colors (see caption of Figure 3). The simulation points that the specimen area is about 25 nm thick, in agreement with the measurement made by convergent beam electron diffraction, which is definitely not thin enough to neglect the dynamical diffraction effects [22]. The other element pointed by the simulation is that the maxima in the experimental image are centered on the Sr and Ti+O be atomic columns, whereas the O columns are invisible in the HRTEM image, even if the image has been acquired at Scherzer’s defocus [22].

This is not surprising, as the HRTEM image contrast for a thick specimen cannot be safely directly interpreted in terms of specimen structure, as it is the result of the interference of the Bloch waves excited in the specimen convoluted with the response of the microscope objective lens [1]. Actually, the comparison of the HRTEM image with the relevant simulation is necessary for understanding the structural information in relationship with the maxima and the minima in the HRTEM image [1]. Furthermore, the intensities in the HRTEM image are not quantitatively correlated with the relevant projected potentials, due also to the dynamical effects that mix the intensities of the different Bloch waves in multiple interactions. In fact, if the dynamical effects could be at least partially amended, it would be possible to achieve a phase retrieved image which is a quantitative map of the projected potential in the specimen. Indeed, in the case shown in Figure 3, by knowing the specimen composition within 10% of accuracy, we estimated the maximum scattering intensity of the structure as the sum of the scattering factors of the atoms in the crystal unit cell [12]. We defined this maximum value as sup{*I*(*s*)}. This is equivalent to consider that each atom scatters in phase with the other atoms in the crystal cell. This is a rough estimation of the effective maximum intensity scattered by the specimen but it is accurate enough for our purposes. Hence, by using the atomic scattering functions, tabulated as a function of the angle [23], we obtain a broad envelope of the maximum scattering amplitude that can be used to rescale the experimental intensities. In fact, this constraint is applied loosely by iterative steps that rescale the experimental intensities to stay behind the envelope curve with a tolerance of about 10% in the intensity rescaling [12]. The results of the approach are shown in Figure 3b, whereas the differences between the raw data and the rescaled intensity are shown in Figure 3c. In Figure 3d there is the comparison, in a logarithmic scale, between the line profile along the dashed blue line in (a) (blue curve) and along the dashed red line in (b) (red curve) after rescaling. The black curve is the profile of the envelope curve calculated according to the estimated maximum scattering amplitude as a function of the angle [12]. The phase reconstruction imaging procedure [14] applied to the rescaled nanodiffraction in Figure 3b is reported in Figure 4. Figure 4a is the phased image and makes immediately evident, in comparison with the crystal structure in Figure 4b and the simulated projected potential in Figure 4c, that in the phase reconstructed image the projected potential of all of the atomic columns are visible and the image is a structural map of the SrTiO_3_ potentials projected on the (001) crystallographic plane. The resolution of the image is related to the finer spatial frequency in the pattern in Figure 3, which is of 65 pm. Figure 4c,d are the simulated and experimental SrTiO_3_ projected potentials, respectively, and the agreement between them, is not only qualitative, but also quantitative in the intensity of each projected potential atomic columns, within an accuracy of 5%. This is a good agreement, if we consider that the crystal potential calculations performed by linear combination of atomic potentials, as those in Figure 4c, are affected by an error of about 10% [24].

The agreement in the width of the experimental and calculated atomic potential fits well with the resolution of 65 pm expected in the phase reconstructed image [12,14]. Finally, it is worth underlining how the partial correction of the dynamical effects in the nanodiffraction patterns enables EDI/KEDI to become methods capable to produce atomic resolution images not only of the spatial projection of the crystal potentials but also of their values, which are directly related to the chemistry of the specimen. This approach results hence in an augmented EDI/KEDI capability of quantitative imaging at atomic resolution.

## 4. Phasing Process

The process of phase reconstruction and the relevant algorithms represent the other pillar on which coherent diffraction imaging has been built up. In the first applications of coherent diffraction imaging in TEM, and up until recently, it was believed that it was mandatory to retrieve the phase, starting from the phase calculated from the FFT of the HRTEM image [5,6,7,9] and that it was not possible to retrieve the phase starting from random phases, as used in coherent diffraction imaging by using X-rays [8]. There were also some debates regarding the need to know the support at a resolution equal to the resolution achievable in the image by phase reconstruction. This would have made EDI/KEDI of little utility from an applicative point of view. Indeed, in a standard EDI/KEDI experiment, the use of a proper optimized phase retrieval algorithm makes it possible to image the projected specimen potential starting from random phases and with a knowledge of the support four times worst with respect to the final resolution achievable in the reconstructed image [14]. The algorithms applied since the beginning of coherent diffraction imaging for the demonstration of the method [8] were mostly derived [25] from the Fienup’s Errors Reduction (ER) and Hybrid Input-Output (HIO) [17], which are an evolution of the Gerchberg–Saxton’s algorithm [18]. ER and HIO are distinguished examples of what is known as deterministic iterative algorithms which, going back and forth from the real to the Fourier’s space, try to minimize a specific error functional [26]. These algorithms, or some of their modifications, are still largely applied nowadays and were also applied in TEM based coherent diffraction imaging up until recently [5,6,7,9]. These algorithms proved to be very efficient in finding local minima of the error functional but, due to the approximate knowledge of the support, they suffer from stagnation. Furthermore, these approaches are also strongly dependent on the choice of the initial conditions related to the tentative random phases [26]. In order to improve the performances and reliability of ER and HIO, and in general of any algorithm based on the minimization of an error functional, usually the phase reconstruction process is repeated several times (thousands) and the better results are averaged to achieve the best estimation of the solution. It is hence evident that the image reconstruction would benefit of a method capable to find a successful strategy to explore accurately the space of the solutions; this is a classic example of what is called Optimization Method [27]. Recently, we proposed a hybrid stochastic method that explores the space of solutions by using a modified Genetic algorithm. This method demonstrates a high efficiency and accuracy in the phase reconstruction with respect to the approaches used so far [14]. This has been demonstrated on both synthetic and experimental KEDI data. The approach is based on a Memetic Algorithm (MA) [28] that merges stochastic and deterministic approaches to explore efficiently the space of solutions. The method has been called Memetic Phase Retrieval (MPR) and in Figure 5 its scheme, in comparison with the standard phasing scheme [14], is reported. The method has been tested on synthetic data in comparison with standard phasing approaches. To this end, the same number of possible solutions within MPR and standard use of HIO/ER, both in case of real 2D data and complex 2D data, were considered. The test demonstrates that MPR works very accurately even in case of loose support. In particular the errors [14] in the reconstruction of 2D real data by using standard phasing methods was of 24%, whereas MPR reached a maximum error of 1%. In the case of complex scattering functions, MPR has been tested considering the knowledge of the support at a resolution four times worst with respect to the resolution expected in the retrieved image. This is a condition normally worst with respect to the condition usually encountered in KEDI experiments where the resolution of the HRTEM is only two or three times worst with respect the spatial resolution of the final reconstruction [9]. In the reconstruction of complex 2D data, standard phasing approaches had an error of 10% whereas MPR reached an error of 1.5% [14]. Hence, MPR accuracy paves the way for quantitative EDI/KEDI results. The configuration of the atomic projected potential is correctly visualized, leading to the direct understanding of the specimen atomic columns distribution and giving definitive information on the structure of interfaces, grain boundaries and defects. Moreover, MPR is also able to retrieve the intensity of the projected potential with enough accuracy to understand the distribution of the different chemical species in the specimen. As a result, the retrieved image is a quantitative atomic resolution map of the distribution of the different atomic species. This opens new perspectives for a future 3D reconstruction of the atomic potential by coherent diffraction imaging. 

In Figure 6 the MPR recovered image of the KEDI experiment, whose HRTEM and nanodiffraction pattern are shown in Figure 3, is reported as example. It is worth noting that this phase-recovered image represents a complex valued scattering function in the direct space. Hence, both its retrieved modulus and phase are shown in the figure. Indeed, in the retrieved image in Figure 6, the brightness is proportional to the modulus while the hue indicates the retrieved phase of the scattering function. By comparing the inset of Figure 6 with the insets within the experimental HRTEM image in Figure 3, it results that the reconstructed image displays the correct crystal projected potentials for SrTiO_3_ (001), showing the atomic columns of oxygen not visible in the experimental HRTEM. From the results in Figure 6 the area shown in Figure 4d has been extracted. The latter is compared with the simulated SrTiO_3_ (001) crystal projected potential shown in Figure 4c. The experimental projected potentials intensities are quantitatively in agreement, within 5%, with what expected by simulating the projected potentials of Sr, Ti+O and O columns projected on the (001) crystallographic plane and hence represent a structural and chemical map of the SrTiO_3_ specimen. The widths of the projected potential peaks are also in agreement with the spatial resolution of 65 pm of the retrieved image, as evidenced by the comparison with the simulated projected potentials in Figure 4c. 

This experiment demonstrates the capability of coherent diffraction imaging in TEM to achieve a resolution of 26λ. Further KEDI experiments on interfaces and defects confirm the capability of MPR to retrieve quantitatively the specimen projected potential also around the interfaces or around the defects, making EDI/KEDI an approach more reliable, flexible and quantitative with respect to past results. 

## 5. Conclusions and Future Perspectives

Coherent diffraction imaging experiments in TEM facilitates accuracy and resolution on standard non-aberration corrected microscopes comparable to those obtained by aberration corrected equipment. The procedures here developed for EDI experiments, when applied to data obtained by aberration corrected equipment, push to the ultimate limit the resolution and the accuracy in the measure of the properties of the matter. The recent results point out the importance of appropriate data reduction and of new more efficient phase retrieval algorithms in EDI/KEDI experiments. The intrinsic feature of an EDI/KEDI experiment, which is based on the acquisition, on the same specimen area, of information in both direct and reciprocal spaces, although from one side it requires to develop a new electron optical tuning of the microscope and the acquisition of both HRTEM and nanodiffraction pattern, on the other side opens new perspectives to maximize the information that can be obtained from the illuminated area. The approach we are following for EDI/KEDI is to develop well controlled procedures capable to originate a retrieved image in which not only the resolution is improved with respect to what is achievable in standard HRTEM experiments due to the residual lens aberrations. In fact, the information we access, in the direct and in the reciprocal space, are merged together in the retrieved image enabling to have as a result a structural and quantitative image, which is a map of the projected potentials at the ultimate resolution.

So far EDI/KEDI experiments were performed by tuning standard microscopes, but it can be envisaged that the features of the EDI/KEDI optimized equipment could ameliorate the coherent diffraction imaging experiment by electron sources with higher coherence, shape optimized condenser apertures, or more sensitive and accurate electron detectors, such as the direct conversion devices recently available to improve the data acquired in the direct space [29], or higher dynamic range detectors to improve the data acquired in the reciprocal space [30].

On the side of the data reduction and of the refinement of the experimental data by a priori information, the results so far obtained encourage the development of further methods that from one side improve the signal to noise ratio in the HRTEM image and nanodiffraction pattern and on the other side make possible to merge efficiently the information both from the direct and the reciprocal space. These features, together with the corrections of the dynamical effects, produce images which are quantitative atomic resolution maps of the specimen potentials enabling to have data that are, also from an intuitive point of view, a display of the atomic species in the sample. This is what we called augmented EDI/KEDI quantitative imaging.

Also, from the point of view of the phasing algorithms, even if the recent result on MPR represents a big leap forward, there is still a large margin of improvement from the point of view of handle noisy data or to overcome the limit of the present algorithms to tackle highly symmetric data, as it happens often in the KEDI experiments.

Important fields of applications of EDI/KEDI are the study of defects in crystals and the study of non-crystalline materials. These are now some of our active areas of research for augmented EDI/KEDI, to achieve quantitative imaging for defects in crystals, hetero-interfaces, multi-phases materials, and amorphous materials. The future perspectives for EDI/KEDI include the 3D reconstruction of the crystal potentials in the illuminated area, or on a single nanoparticle, and the possibility for effective imaging in time resolved TEM experiments. Finally, the synergy between EDI/KEDI and aberration corrected experiments is yet to be explored in detail and it can be seen that the combination of the two approaches could be used in the future to solve challenging problems at the highest spatial resolution and accuracy.

## Figures and Tables

**Figure 1 materials-11-02323-f001:**
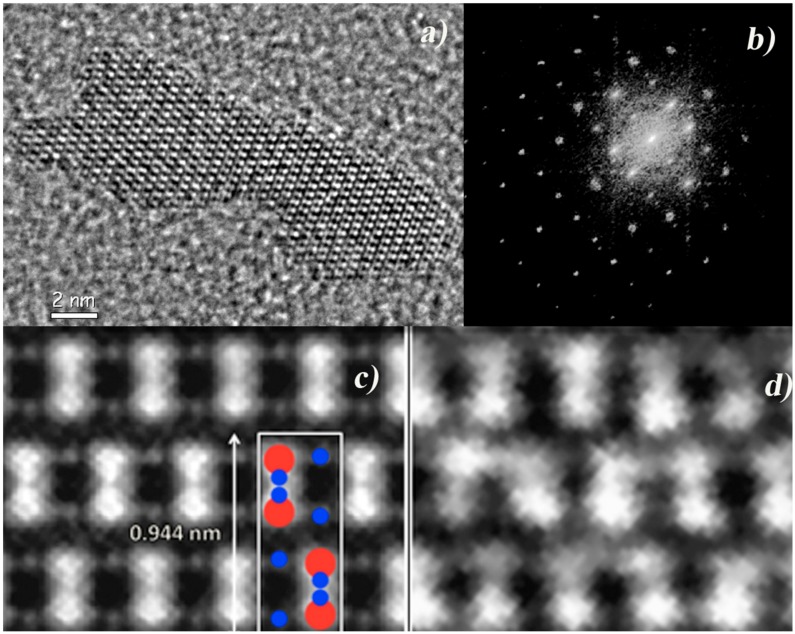
(**a**) HRTEM image of a TiO_2_ nanorod in [100] zone axis; (**b**) nanodiffraction pattern complemented with the Fast Fourier Transform (FFT) of the HRTEM after rotation and scaling; (**c**) image after phase reconstruction; (**d**) magnified view of a portion of the HRTEM image in (**a**). (Reprinted with permission from De Caro et al. [7]).

**Figure 2 materials-11-02323-f002:**
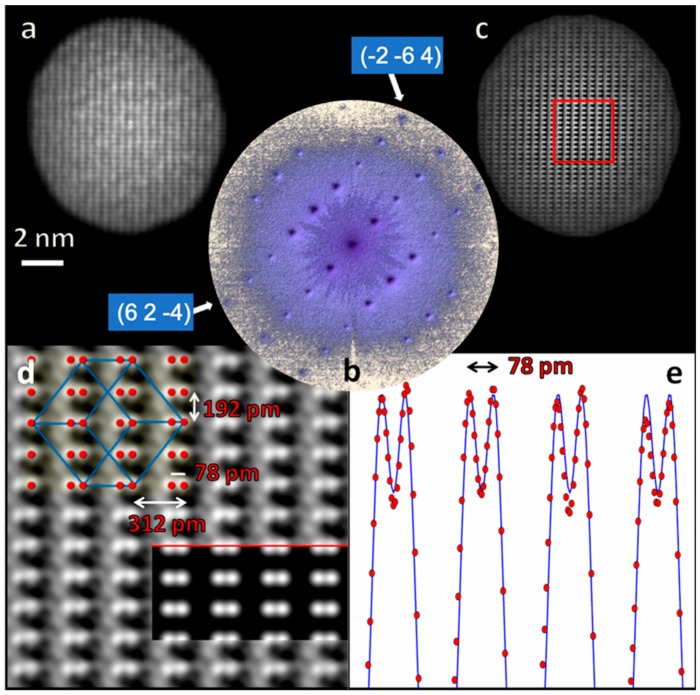
Keyhole electron diffraction imaging (KEDI) experiment on Si [112]: (**a**) HRTEM imaged of the self-confined illuminated area (support); (**b**) nanodiffraction pattern from the area shown in the HRTEM experiment in (**a**), the arrows point the diffracted spots at the highest frequency corresponding to a lattice spacing of 72 pm; (**c**) image reconstruction in the illuminated area after phase retrieval process; (**d**) magnified view of the image in (**c**) together with the atomic columns position within the Si crystal cell in [112] projection. The inset shows the simulation of the atomic projected potential in [112] projection; (**e**) simulated (dots) and measured (solid line) intensity profile of the Si atomic columns in [112] projection showing the Si dumbbell spacing at 78 pm well resolved.

**Figure 3 materials-11-02323-f003:**
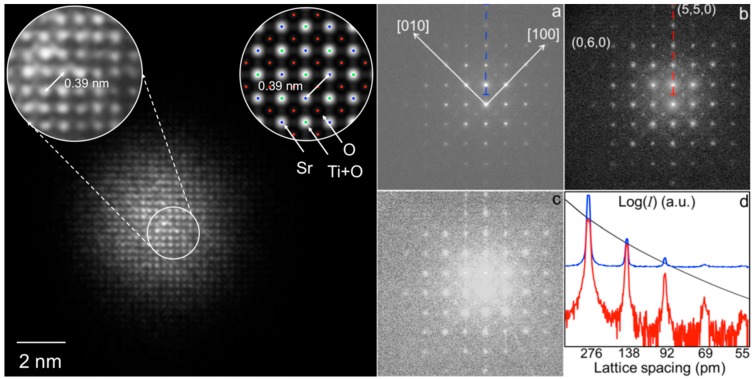
Left- HRTEM image of a nano-region of a SrTiO_3_ extended sample in [001] zone axis, with a zoom in the left inset, and the relevant simulation in the right inset (objective lens underfocus of 41.3 nm and specimen thickness of 25.0 nm); the dots in the simulation point to the structural positions of the SrTiO_3_ atomic species in [001] projection: Sr = Blue, Ti+O = green, O = red. (**a**) KEDI raw experimental nano-diffraction pattern (logarithmic scale); (**b**) sup{*I*(*s*)} rescaled pattern; (**c**) difference between patterns shown in (**a**,**b**); (**d**) comparison in a logarithmic scale between the line profile along the dashed blue line in (**a**) (blue curve) and along the dashed red line in (**b**) (red curve) after rescaling. Black curve is the corresponding profile of sup{*I*(*s*)} × *I*_max_ constraint. (Reprinted with permission from De Caro et al. [12]).

**Figure 4 materials-11-02323-f004:**
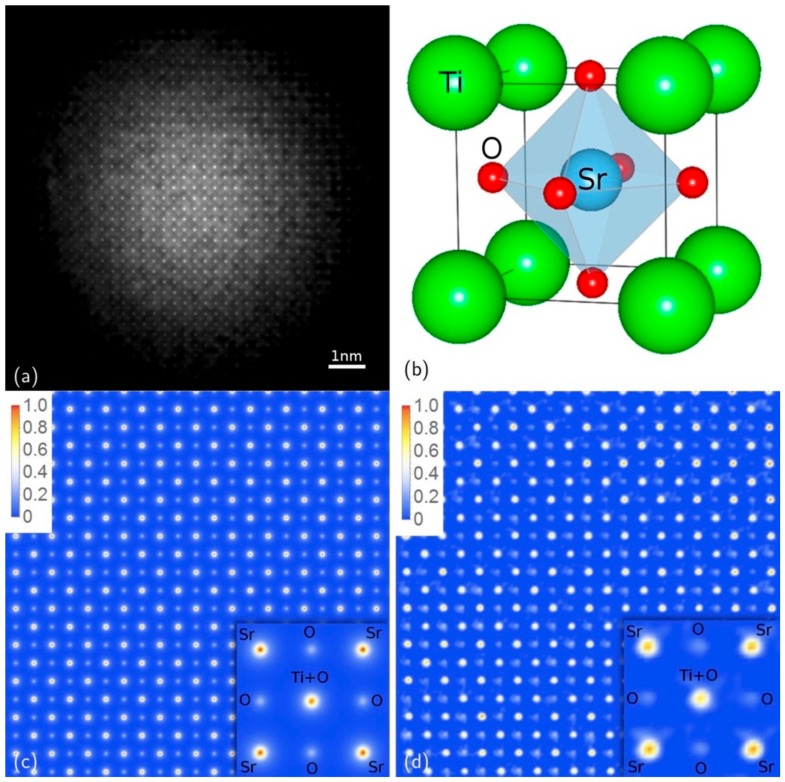
(**a**) Modulus of the retrieved scattering function (relevant to Figure 3); (**b**) SrTiO_3_ unit cell; (**c**) simulation of the SrTiO_3_ projected potential in [001] zone axis and (**d**) experimental data extracted from the phased map. (Reprinted with permission from Colombo et al. [14]).

**Figure 5 materials-11-02323-f005:**
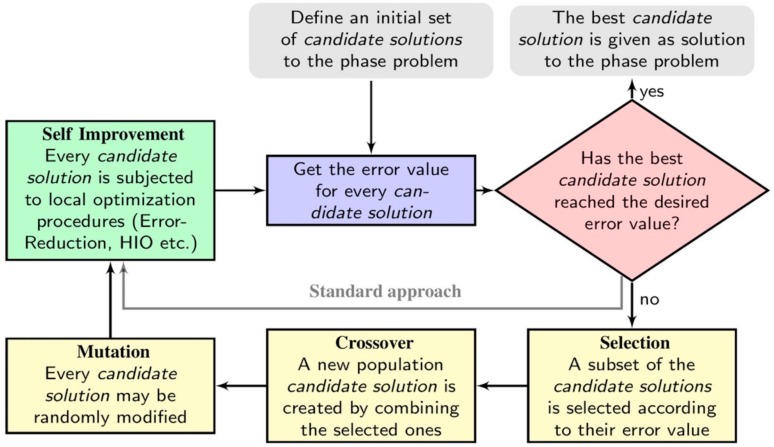
Description of the Memetic Phase Retrieval (MPR) approach. The standard approach can be interpreted as MPR deprived of genetic operations of Selection, Crossover and Mutation.

**Figure 6 materials-11-02323-f006:**
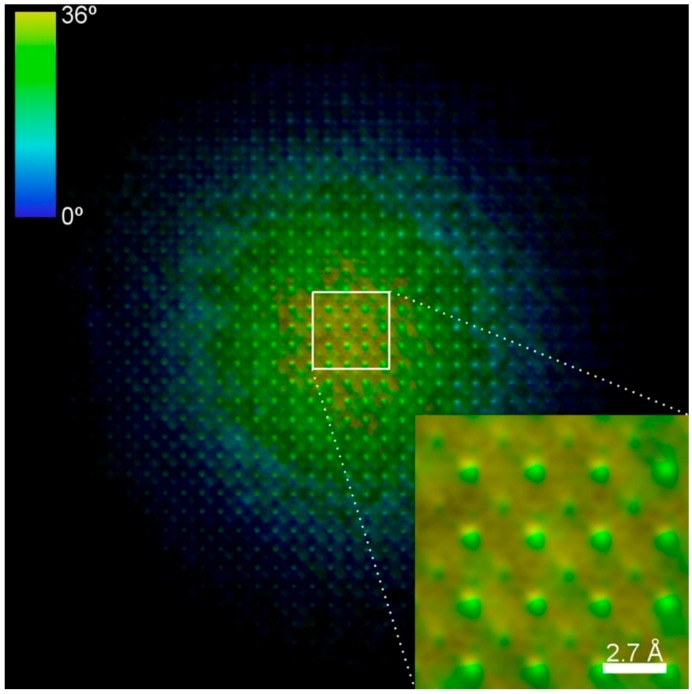
Retrieved image by MPR for the KEDI experiment in Figure 3. Here the brightness is proportional to the retrieved modulus whereas the colors represent the retrieved phases. (Reprinted with permission from Colombo et al. [14]).

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
