# Peer review of "Coherent Diffraction Imaging in Transmission Electron Microscopy for Atomic Resolution Quantitative Studies of the Matter"

_materials, 2018, doi:10.3390/ma11112323_

Reviewer 1 Report

This paper reviews recent achievements using coherent diffraction imaging in TEM for atomic resolution microscopy of crystalline materials. The paper is nicely written and detailed discussions of previously performed studies are reported.

Nevertheless, minor editing is suggested to improve the article:

1) line 85: 'nanodiffraction patterns' instead of 'nanodiffractions';

2) line 133: 'Coulomb potential of the sample medium';

3) line 134: 'plane wave illumination' instead of 'even illumination';

4) line 138: 'higher spatial frequency';

5) Figures 2, 4 and 6 appear as oversized, and could be reduced in size by a factor of 2;

6) lines 261, 262: 'dynamical diffraction component' and 'dynamical diffraction patterns' are to be used instead of 'dynamicalcomponent' and 'dynamical patterns', respectively;

7) Figure 3: the blue, pale blue and pale green dot colours are neither visible nor distinguishable in the right inset of left image. Using different symbols can help to improve this;

8) lines 294, 300: 'Bloch waves' instead of 'Block's waves';

9) line 301: 'amended' instead of 'emended';

10) In Conclusions it would be helpful to discuss the perspectives of the method to study non-crystalline materials.

Author Response

Answer to reviewer 1.

(R)This paper reviews recent achievements using coherent diffraction imaging in

TEM for atomic resolution microscopy of crystalline materials. The paper is

nicely written and detailed discussions of previously performed studies are

reported.

(A) As first we would like to thank reviewer 1 for his/her positive opinion on our work and for the suggestions to improve our paper.

(R) 1) line 85: 'nanodiffraction patterns' instead of 'nanodiffractions';

2) line 133: 'Coulomb potential of the sample medium';

3) line 134: 'plane wave illumination' instead of 'even illumination';

4) line 138: 'higher spatial frequency';

5) Figures 2, 4 and 6 appear as oversized, and could be reduced in size by a

factor of 2;

6) lines 261, 262: 'dynamical diffraction component' and 'dynamical diffraction

patterns' are to be used instead of 'dynamical component' and 'dynamical

patterns', respectively;

7) Figure 3: the blue, pale blue and pale green dot colours are neither visible

nor distinguishable in the right inset of left image. Using different symbols can

help to improve this;

8) lines 294, 300: 'Bloch waves' instead of 'Block's waves';

9) line 301: 'amended' instead of 'emended';

(A) The paper has been emended according to all his/her suggestions, as marked in the paper. In particular we replaced figure 3 with a new one, where we used more evident colors for the right inset of the left image. The relevant caption has been emended accordingly.

(R) 10) In Conclusions it would be helpful to discuss the perspectives of the

method to study non-crystalline materials.

(A) In Conclusionswe added the following statements:

Important fields of applications of EDI/KEDI are the study of defects in crystals and the study of non-crystalline materials. These are now some of our active areas of research for augmented EDI/KEDI, to achieve quantitative imaging for defects in crystals, hetero-interfaces, multi-phases materials, and amorphous materials.

Reviewer 2 Report

Materials

The paper reviews the technique of coherent electron diffraction imaging that allows obtaining very high resolution potential maps without needing  expensive aberration corrected instruments.  I think the paper is well written, the review is useful and covers all importantt bases, appears scientifically solid and can be recommended for publication with minor revision, mostly related to clarity.

1. The following sentences are not understandable, or something is missing:

"If f is the smallest frequency measurable in the diffraction pattern, hence sf ≥ 1. "

"The basic data reduction for nanodiffractions includes background subtraction, noise reduction and eventual masking whereas, for the relevant HRTEM images, it is necessary an accurate process of scaling and rotation in order to have the relevant FFT that matches precisely the nanodiffraction pattern. "  Probably "to apply"is missing.

" The intrinsic feature of an EDI/KEDI experiment, which is based on the acquisition, on the same specimen area, of information in both direct and reciprocal spaces, if from one side requires to develop a new  electron optical tuning of the microscope and the acquisition of both HRTEM and nanodiffraction, on the other side opens new perspectives to maximize the information that can be obtained from the illuminated area. "  Probably, the   "if" should be replaced by something else

2. I think the authors mean specific instead of peculiar in the following fragment:      "Figure 1 shows the peculiar steps of an EDI experiment: ", p.4

3. The text should be checked for many small languagemistakes.

4. The authors frequently use the term nanodiffractions instead of nanodiffraction      patterns. In my opinion, this is not correct.

5. Figure 3  "the dots in the simulation point to the structural positions of the      SrTiO3 atomic species in [001] projection: Sr = Blue, Ti+O = pale blue, O = pale green" - I suppose the dots are the little points in the right      side circle superimposed on the atom columns, but these are too small to      discriminate any colors

Author Response

Answer to reviewer 2.

(R) The paper reviews the technique of coherent electron diffraction imaging

that allows obtaining very high resolution potential maps without needing

expensive aberration corrected instruments. I think the paper is well

written, the review is useful and covers all important bases, appears

scientifically solid and can be recommended for publication with minor

revision, mostly related to clarity.

(A) As first we would like to thank reviewer 2 for his/her positive opinion on our work and for the suggestions to improve our paper. In the following the answer to the point raised by the reviewer:

(R) The following sentences are not understandable, or something is

missing:

"If f is the smallest frequency measurable in the diffraction pattern, hence

sf ≥ 1. "

(A) In the paper we define S as the size of the support but in (sf ≥ 1) is reported in lower case and this likely produced a misunderstanding. According to the Shannon’s theorem the minimum frequency to sample the Fourier transform of the support in the reciprocal space is “S” and hence if “f” is the maximum spatial frequency experimentally measured,hence Sf has to be at least 1 or, in case of oversampling, Sf>1.

We changed the case of “s” in the emended paper.

(R) "The basic data reduction for nanodiffractions includes background

subtraction, noise reduction and eventual masking whereas, for the

relevant HRTEM images, it is necessary an accurate process of scaling and

rotation in order to have the relevant FFT that matches precisely the

nanodiffraction pattern. " Probably "to apply" is missing.

(A) The paper has been emended accordingly

(R) " The intrinsic feature of an EDI/KEDI experiment, which is based on the

acquisition, on the same specimen area, of information in both direct and

reciprocal spaces, if from one side requires to develop a new electron

optical tuning of the microscope and the acquisition of both HRTEM and

nanodiffraction, on the other side opens new perspectives to maximize

the information that can be obtained from the illuminated area. "

Probably, the "if" should be replaced by something else

(A) The use of this phrase construction, a kind of adversative phrase with “if”, is used to mark, to underline, that from one side the experiment is not standard and requires the use of non-standard illumination set up which complicates the experiment, but this extra effort produces several advantages in terms of information that can be obtained from the specimen. We replaced “if” with “although”.

(R) 2. I think the authors mean specific instead of peculiar in the following

fragment: "Figure 1 shows the peculiar steps of an EDI experiment: ",

p.4

(A) We use “peculiar” in the sense described by the oxford dictionary and reported below:

distinctivecharacteristic, distinct, individual, special, idiosyncratic, unique, personal.”

(R) 3. The text should be checked for many small languagemistakes

(A) the paper has been checked accordingly.

(R) 4. The authors frequently use the term nanodiffracttions instead of

nanodiffraction patterns. In my opinion, this is not correct.

(A) the paper has been emended accordingly

(R) 5. Figure 3 "the dots in the simulation point to the structural positions

of the SrTiO3 atomic species in [001] projection: Sr = Blue, Ti+O = pale

blue, O = pale green" - I suppose the dots are the little points in the

right side circle superimposed on the atom columns, but these are too

small to discriminate any colors.

(A) We replaced figure 3 with a new one with more visible colours

Reviewer 3 Report

Dear editor dear authors,

the manuscript „coherent diffraction imaging in transmission electron microscopy for atomic resolution quantitative studies of the matter“ gives an review of the author's work in recent years on the before mentioned topic. Paragraph 1 gives an introduction to the field, followed by the description of the experimental setup and the background for the possibilty of phase reconstruction in terms of the Shannon and the generalized sampling theorem. Paragraph 3 focuses on the data reduction and introduces a new technique called augmented EDI/KEDI. Finally, different phasing algorithms are discussed (paragraph 4) and future perspectives are given.

In summary, I think it is a successful review of the author's work and therefore I recommend publication after some minor revisions.

In the abstract: „The paper focuses the attention on...“ Maybe a bit of philosophical remark: I think it is not clear, whether the paper will focus the attention of the community to EDI. Hopefully, it will, but still I would just write → „The paper focuses on ...“

Page 3: „The success and the accuracy“ → Aren't success and accuracy the same? The reconstruction is only successfull if it is accurate?

Eq. 2: „k“ should be vectors

Page 5 on the topic of KEDI: Here I think the authors could explain in more detail, how the the defined illuminated area is achieved exactly. Small condensor aperture? Nanoprobe mode?

Page 5: What is the advantage of KEDI over EDI? Is it not possible to reconstruct a crystal defect in EDI?

Fig. 2d and e: Can the authors shortly discuss where the artificial intensity in between the atom positions comes from. Then the intensity profiles in e need not to be cut

Intensities in a diffractogram (FFT of an image) and intensities from a diffraction pattern are not the same. How does this influence the result of the reconstruction ?

page 8: „Block' s waves“ → „Bloch waves“

page 8: „Fig.3“ → „Fig. 3“

page 8: About the usage of atomic scattering functions for the estimation for estimation of the envelope: The scattering amplitudes used are for isolated atoms, which neglect the redistribution of charge due to bonds. However, the brightest spots are most severly influenced by the redistribution and exactly these intensities are used to estimate the „amplitude“ of the envelope, that is used to later scale the experimental dynamic intensities to „quasi-kinematic“ instensities. Does that make an influence?

Does EDI in the end really help to improve resolution for corrected TEMs ? If so, why since diffraction patterns are not influenced by aberrations of the objective lens, but might be stronger distorted due to the corrector. What are the perspectives for such machines?

Page 13: Also the higher dynamic range of new detectors can be of help for the technique (e.g. Medipix, just to mention the detector with a large dynamic range and with the most pixels (512x512) available at the moment (at least to my knowledge))

Author Response

Answer to reviewer 3.

(R) the manuscript „coherent diffraction imaging in transmission electron

microscopy for atomic resolution quantitative studies of the matter“ gives an

review of the author's work in recent years on the before mentioned topic.

Paragraph 1 gives an introduction to the field, followed by the description of the

experimental setup and the background for the possibilty of phase

reconstruction in terms of the Shannon and the generalized sampling theorem.

Paragraph 3 focuses on the data reduction and introduces a new technique

called augmented EDI/KEDI. Finally, different phasing algorithms are

discussed (paragraph 4) and future perspectives are given.

In summary, I think it is a successful review of the author's work and therefore I

recommend publication after some minor revisions. 

(A) As first we would like to thank reviewer 3 for his/her positive opinion on our work and for the suggestions to improve our paper.

(R) In the abstract: „The paper focuses the attention on...“ Maybe a bit of

philosophical remark: I think it is not clear, whether the paper will focus the

attention of the community to EDI. Hopefully, it will, but still I would just write →

„The paper focuses on ...“

(A) the paper has been emended accordingly

(R) Page 3: „The success and the accuracy“ → Aren't success and accuracy the

same? The reconstruction is only successfull if it is accurate?

(A) Definitely the reconstruction is successful if it is accurate but the precision in the reconstruction also depends on the robustness of the available information. Hence, we emended the phrase as follow: The accuracy and the precision……

(R) Eq. 2: „k“ should be vectors

(A) the paper has been emended accordingly

(R) Page 5 on the topic of KEDI: Here I think the authors could explain in more

detail, how the the defined illuminated area is achieved exactly. Small

condensor aperture? Nanoprobe mode?

(A) We add the following statement to the emended paper:

The cathode emission condition and the electron optical illumination system of the microscope have to be set up in free lens control to increase the probe coherence on the smallest illuminated area achievable. The microscope has to be operated starting from a standard nano-probe configuration with the smallest condenser aperture, typically a condenser aperture of 10 micron. The three magnetic lenses of the illumination system of the microscope have to be operated independently, together with the electrostatic lens of the emitter, to produce the smallest probe on the focal plane of the pre-field of the objective lens to obtain the smallest coherent parallel beam on the specimen. To further increase the coherence of the electron-probe, the heating current has to be decreased with respect to the heating current used in standard TEM imaging. Typically, this cathode set-up results in a reduction of the emission current of about 50% in a probe which has a minimum size on the specimen between 5 nm to 10 nm. The current density on the specimen is below the detection limit of the microscope phosphorus screen (<0.1 pA cm-2). This set up enables to acquire the diffraction pattern without using the beam stopper. This is an immediate advantage with respect to the experimental set up used in EDI where the use of the beam stopper makes the diffraction pattern incomplete. In KEDI experiments, to……..

(R) Page 5: What is the advantage of KEDI over EDI? Is it not possible to

reconstruct a crystal defect in EDI?

(A): The advantage of KEDI is mainly to enable the study of nanometric areas of extended specimen and hence it can be used, for example, to study defects, interfaces and amorphous materials. This is an area of research on which we are now active and we have promising preliminary results. We added a statement on this subject in the Conclusion of the emended paper:

Important fields of applications of EDI/KEDI are the study of defects in crystals and the study of non-crystalline materials. These are now some of our active areas of research for augmented EDI/KEDI, to achieve quantitative imaging for defects in crystals, hetero-interfaces, multi-phases materials, and amorphous materials.

 For your convenience we enclose a confidential image, not yet published, where you can see the differences between standard HRTEM and KEDI to image an interface.

(R) Fig. 2d and e: Can the authors shortly discuss where the artificial intensity in

between the atom positions comes from. Then the intensity profiles in e need

not to be cut

(A) We added the following statement to the emended paper:

Some small artifacts at low intensities, in between the projected atomic columns, are visible in Fig. 2d. These artifacts are similar to those shown in Fig. 6 of ref. [9] and could be related to small dynamical diffraction contributions to the diffracted intensities.

(R) Intensities in a diffractogram (FFT of an image) and intensities from a

diffraction pattern are not the same. How does this influence the result of the

reconstruction ?

(A) This is part of the necessary data reduction for EDI/KEDI: the intensities at relatively low spatial frequencies in the FFT are matched in intensity and geometry with the relevant intensities in the experimental diffraction pattern before merging the two patterns. The resulting pattern is the starting point for the phase retrieval process. This is reported at page 7 and widely discussed in the relevant references.

(R) page 8: „Block' s waves“ → „Bloch waves“

(A) The paper has been emended accordingly

(R) page 8: „Fig.3“ → „Fig. 3“

(A) The paper has been emended accordingly

(R) page 8: About the usage of atomic scattering functions for the estimation for

estimation of the envelope: The scattering amplitudes used are for isolated

atoms, which neglect the redistribution of charge due to bonds. However, the

brightest spots are most severly influenced by the redistribution and exactly

these intensities are used to estimate the „amplitude“ of the envelope, that is

used to later scale the experimental dynamic intensities to „quasi-kinematic“

instensities. Does that make an influence?

(A) The constraint we use starting from the atomic scattering functions is applied broadly to the experimental intensities. This enabled to scale partially the effect of dynamical scattering but leaving the solution to the reconstruction to have enough degrees of freedom to describe the peculiarity of the coulomb potential of the specimen under study. The charge redistribution definitely has an influence and in some of the experiments we performed there are some clues that indicate this influence. The problem is how to distinguish between the charge redistribution effect and eventual artifacts of the image reconstruction. This is one of the main problem on which we are working now. Preliminary results are encouraging and we expect to find a reliable way to manage this problem in the near future.

(R) Does EDI in the end really help to improve resolution for corrected TEMs ? If so, why since diffraction patterns are not influenced by aberrations of the

objective lens, but might be stronger distorted due to the corrector. What are

the perspectives for such machines?

(A) Coherent diffraction imaging demonstrated good performances in retrieving the phase, and the phase carries detailed information on the image that could ameliorate the results of standard HRTEM in aberration corrected TEM. Furthermore, HRTEM image is an interference pattern and the structural information should be extracted and compared to a model for reliable measurements. Augmented EDI/KEDI aim to produce a structural quantitate map of the specimen.

Coherent diffraction imaging in an aberration corrected TEM is an aspect that nobody explored so far, to our knowledge, and it is a technological and scientific opportunity that should be understood and deepened. The quality of EDI/KEDI results is a combination of the information available in the direct space (called a priori information in analogy to coherent diffraction imaging with x-ray) and in the reciprocal space. For example, the knowledge of the support is a common example of a priori information used in coherent diffraction imaging. The precision in the knowledge of the support is expected to be better by using aberration corrected HRTEM with respect to conventional microscopes. In general, all the information a priory known on the specimen can contribute to a higher precision and accuracy in the phase reconstruction. This is the reason of Augmented EDI/KEDI method. We believe that would have been synergic advantaged in EDI/KEDI experiments in aberration corrected equipment. But, of course, this has to be proved.

At the same time, it is certainly necessary to this aim to find a reliable way to correct the distortion induced by the correctors in the diffraction patterns. As always, the reliability of a TEM experiment depends on the control and the understanding of the conditions used during the experiments.

(R) Page 13: Also the higher dynamic range of new detectors can be of help for

the technique (e.g. Medipix, just to mention the detector with a large dynamic

range and with the most pixels (512x512) available at the moment (at least to

my knowledge))

(A) We reported this suggestion in the emended paper with the relevant reference to the Medipix Collaboration and we added also a mention to direct conversion detector for better direct space information with the relevant reference. 

Reviewer 4 Report

The authors applied coherent electron diffraction imaging to study materials properties at atomic resolution. The improved EDI can overcome aberrations by experimental set-up, data processing and algorithms. The manuscript is well written, the method is informative to the readers, I would suggest the paper be accepted after minor revisions; the following concerns should be noticed.

In the first paragraph of the main text, line 40-50, it is advised that the first few sentences be refined, as they come from the same reference [1]. 

In EDI experimental part, it's recommended that a diagram of the system is provided to illustrate the setup. 

The editor, however, might need to ensure that this review fits aim and scope of the journal. 

Author Response

Answer to reviewer 4.

(R) The authors applied coherent electron diffraction imaging to study materials

properties at atomic resolution. The improved EDI can overcome aberrations

by experimental set-up, data processing and algorithms. The manuscript is

well written, the method is informative to the readers, I would suggest the

paper be accepted after minor revisions; the following concerns should be

noticed.

(A) As first we would like to thank reviewer 4 for his/her positive opinion on our work and for the suggestions to improve our paper.

(R) In the first paragraph of the main text, line 40-50, it is advised that the first few

sentences be refined, as they come from the same reference [1].

(A) We agree that the same reference is cited several consecutive times but the concepts expressed in the relevant statements could be straightforward for readers expert in TEM and the relevant reference could be just reported for them at the end as well-known. For scientists not expert in TEM we would like to make clear that each of the fundamental concepts can be deepen in the book in ref [1].

(R) In EDI experimental part, it's recommended that a diagram of the system is

provided to illustrate the setup.

(A) We added in the par. 2 of the emended paper a note that refers to the graphical abstract where the scheme of a coherent diffraction imaging experiment in a TEM is reported with the electron optical configuration in imaging and diffraction. We also added in par. 2 some details on the used experimental configuration for KEDI:

The cathode emission condition and the electron optical illumination system of the microscope have to be set up in free lens control to increase the probe coherence on the smallest illuminated area achievable. The microscope has to be operated starting from a standard nano-probe configuration with the smallest condenser aperture, typically a condenser aperture of 10 micron. The three magnetic lenses of the illumination system of the microscope have to be operated independently, together with the electrostatic lens of the emitter, to produce the smallest probe on the focal plane of the pre-field of the objective lens to obtain the smallest coherent parallel beam on the specimen. To further increase the coherence of the electron-probe the heating current has to be decreased with respect to the heating current used in standard TEM imaging. Typically, this cathode set-up results in a reduction of the emission current of about 50% in a probe which has a minimum size on the specimen between 5 nm to 10 nm. The current density on the specimen is below the detection limit of the microscope phosphorus screen  (<0.1 pA cm-2). This set up enables to acquire the diffraction pattern without using the beam stopper. This is an immediate advantage with respect to the experimental set up used in EDI where the use of the beam stopper makes the diffraction pattern incomplete. In KEDI experiments, to……..
